# Structural basis of measles virus polymerase inhibition by nonnucleoside inhibitor ERDRP-0519

Dong Wang[1,3], Fan Bu[1,2,3], Ge Yang[1] & Bin Liu[1] ✉

ERDRP-0519 is a potent nonnucleoside inhibitor active against measles virus (MeV) and other Morbilliviruses. Here we report cryo-EM structures of the compound bound to MeV polymerase complexes at 2.73 Å and 2.48 Å resolution, revealing a unique binding pocket in the RdRp palm subdomain that overlaps the catalytic GDN motif. These findings clarify the basis of resistance mutations, including W671, and provide a foundation for designing next-generation Paramyxovirus antivirals.

Paramyxoviruses, a family of negative-stranded RNA viruses, include significant human pathogens such as measles virus (MeV), mumps virus (MuV), human parainfluenza viruses (HPIVs), and Nipah virus (NiV)[1]. MeV is particularly concerning due to its high infectivity and potential to cause severe complications, including pneumonia and encephalitis, especially in infants and young children. Measles was declared eliminated in the U.S. in 2000 following widespread MMR vaccination; however, declining vaccination rates have led to resurgences. In 2025 alone, 14 outbreaks and over 1000 cases were reported, often linked to unvaccinated communities and international travel[2]. Currently, no FDA-approved antiviral treatment exists for MeV, and ribavirin, the only antiviral option for Paramyxovirus, offers limited efficacy and is rarely used for measles[3,4]. This highlights the urgent need for more effective antiviral therapies.

Two small-molecule antivirals, AS-48[5] and ERDRP-0519[6], were developed to target different viral components of MeV: the fusion (F) protein and the L-P polymerase complex, respectively. AS-48 emerged from structure-based design, while ERDRP-0519 was identified through high-throughput screening, with both compounds undergoing multiple rounds of optimization to improve toxicity, solubility and potency[5-8] (Fig. 1a). The F protein, in conjunction with hemagglutinin, facilitates viral entry into host cells, whereas the L-P complex is essential for viral replication. Although AS-48 inhibits the F protein, resistance has emerged, including a resistant MeV isolate from Sub-Saharan Africa and rapid escape mutants observed in vitro[9]. Structural studies have elucidated the mechanism of resistance, leading to reduced prioritization of AS-48 development[10]. Conversely, polymerase inhibitors like ERDRP-0519 show greater promise due to the high conservation of the L-P complex across MeV strains. ERDRP-0519,

a chemically optimized polymerase inhibitor, has demonstrated strong activity against multiple MeV isolates, with evidence of broader efficacy against other Morbilliviruses, positioning it as a leading therapeutic candidate[11]. However, despite its potential, structural information on ERDRP-0519's binding to the L-P complex remains elusive, constraining further optimization efforts (Fig. 1b).

In this work, we determine cryo-EM structures of the MeV polymerase complexes bound to ERDRP-0519 at 2.73 Å and 2.48 Å resolution, revealing a unique binding pocket overlapping the RdRp catalytic GDN motif. Furthermore, cell-based assays demonstrate that a critical hydrogen bond with W671 plays a key role in mediating drug resistance.

## Results and discussion

### Structural basis of ERDRP-0519 bound to MeV polymerase complexes

Previous resistance profiling identified the L protein as the target of ERDRP-0519, with resistance mutations clustering around the conserved GDN motif in the RNA-dependent RNA polymerase (RdRp) and adjacent PRNTase domains[12]. A truncated MeV L variant (L1708), containing only those two domains, yielded a dissociation constant of 78-140 nM for ERDRP-0519, indicating strong binding affinity[13]. Docking analysis predicted that ERDRP-0519 binds between the RdRp and PRNTase domains, near PRNTase motifs A and D[13]. To test this, we assembled the MeV ternary polymerase complex[14] with ERDRP-0519 and analyzed it by cryo-EM. The maps revealed two complexes: $L_{core}$-P and $L_{full}$-P-C, with clear ERDRP-0519 density on the L protein (Fig. 1c, Supplementary Fig. 1, and Supplementary Table 1). Contrary to predictions, ERDRP-0519 was observed binding within the RdRp domain

[1]Section of Transcription & Gene Regulation, The Hormel Institute, University of Minnesota, Austin, MN, USA. [2]Department of Pharmacology, University of Minnesota Medical School, Minneapolis, MN, USA. [3]These authors contributed equally: Dong Wang, Fan Bu. ✉e-mail: liu00794@umn.edu

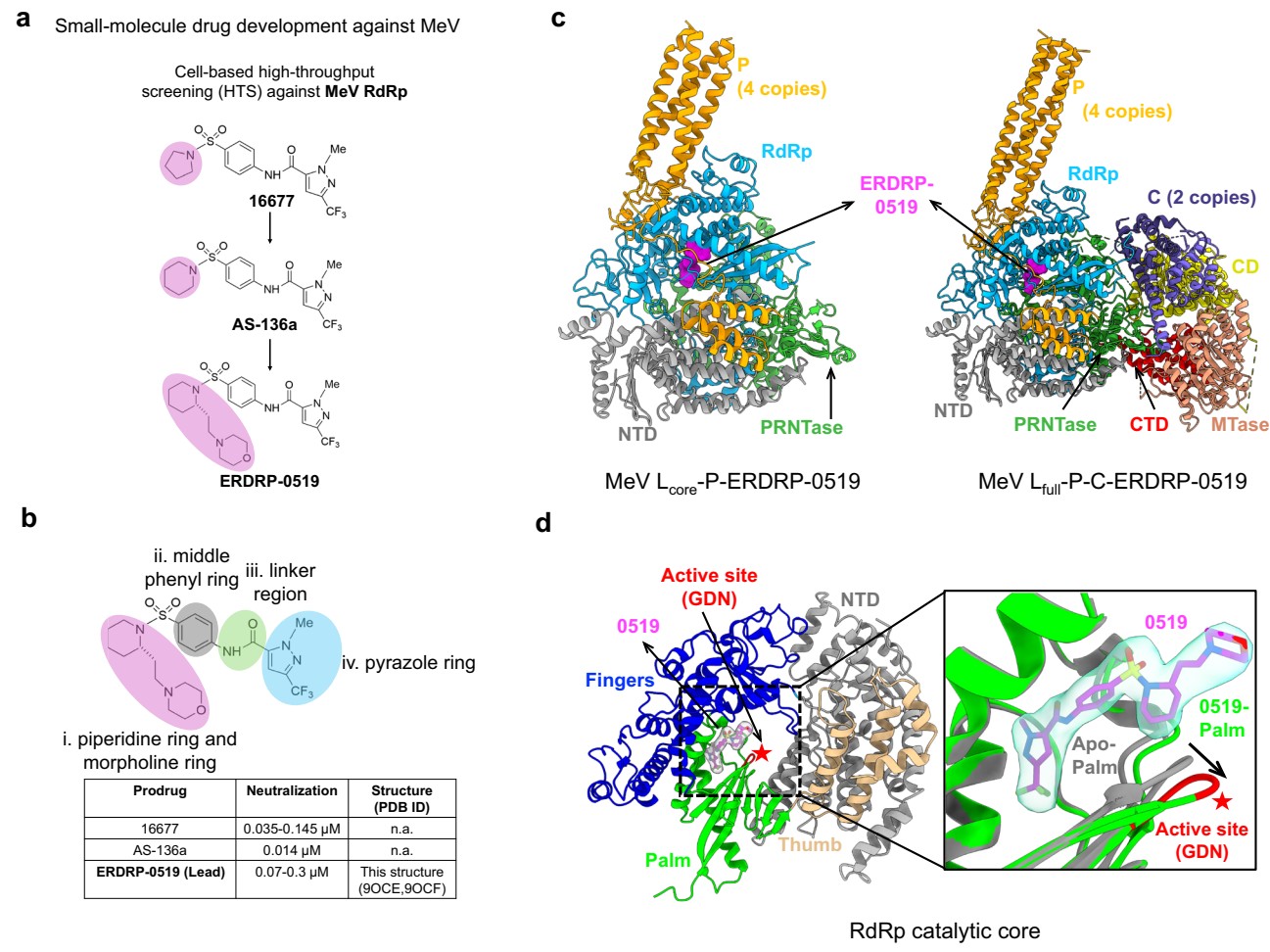

**Fig. 1 | Compound optimization and overall structures. a** Development of small-molecule inhibitors targeting MeV. **b** Chemical structure of the MeV RdRp lead compound ERDRP-0519 and its neutralization potency compared to other prodrugs. **c** Structures of the MeV polymerase complex with ERDRP-0519. The MeV L$_{core}$-P (left) and L$_{full}$-P-C (right) complexes are shown in cartoons with different subunit colors and ERDRP-0519 as magenta spheres. **d** The RdRp catalytic core is shown in cartoons: NTD (gray), fingers (blue), palm (lime), thumb (burlywood), and active site (red). ERDRP-0519 is represented as magenta sticks with transparent gray density. Zoom-in: Superimposition of the palm subdomain of the MeV L$_{full}$-P-C-ERDRP-0519 complex (lime) and apo-MeV L$_{full}$-P-C (gray). The active site loop (GDN) shifts upon compound binding (black arrow).

---

rather than at the predicted PRNTase site (Fig. 1d). Additionally, C protein presence did not alter the binding mode, suggesting stable drug association across RNA synthesis states.

## Molecular interactions and mechanism of ERDRP-0519 inhibition

Structural analysis revealed that ERDRP-0519 binds to the palm subdomain of the RdRp, targeting the catalytic GDN motif and a nearby hydrophobic channel (Figs. 1d, 2a). This binding mode differs from FDA-approved nucleoside RdRp inhibitors, such as Sofosbuvir for hepatitis C virus (HCV)[15] and Ribavirin for foot-and-mouth disease virus (FMDV)[16], which simply occupy the central polymerase cavity (Supplementary Fig. 2). Our data show the binding pocket is framed by escape mutations around the GDN motif, consistent with the resistance observed in the L-T751I mutant, which exhibited a 20-fold increase in EC90 values[11]. Mutations in the PRNTase domain also affect drug binding[13], likely due to their proximity to the central polymerase cavity. The morpholine ring of ERDRP-0519 extends into this cavity, suggesting resistance may arise from structural changes that indirectly influence binding.

Our structures also explain the observation that modifications to the central ring or pyrazole moiety of ERDRP-0519 reduce its antiviral potency[8] (Fig. 2a, b). The central ring forms strong hydrophobic interactions with residue Y667, so any changes at this position are likely to cause steric clashes, disrupting binding. Additionally, the trifluoromethylated pyrazole group fits precisely into a hydrophobic pocket, engaging six surrounding residues and forming more interactions than any other part of the molecule (Fig. 2b). Altering or removing this moiety disrupts these interactions, markedly reducing binding and potency. In contrast, modifications to the piperidine ring are better tolerated, likely because extending it allows the compound to project into the central polymerase cavity, which has enough space to accommodate additional groups. Incorporating a morpholine ring into this region did not create new interactions with the L protein but instead occupied the solvent-accessible channel, partially explaining ERDRP-0519's superior pharmacokinetic profile compared to AS-136A.

Notably, our structural data reveals two key mechanisms by which ERDRP-0519 exerts antiviral activity. First, it inhibits the catalytic center by interacting with residue D773 in the GDN motif (Fig. 2c). The piperidine group displaces D773, creating space for hydrophobic interaction that directly impairs active site function. It is noted that ERDRP-0519 does not affect the viral transcription gradient[13], suggesting that residue D773 may be positioned differently when the polymerase is in an elongation conformation and/or that ERDRP-0519 cannot bind an elongating polymerase. Second, although the morpholine ring does not directly contact the L protein, structural

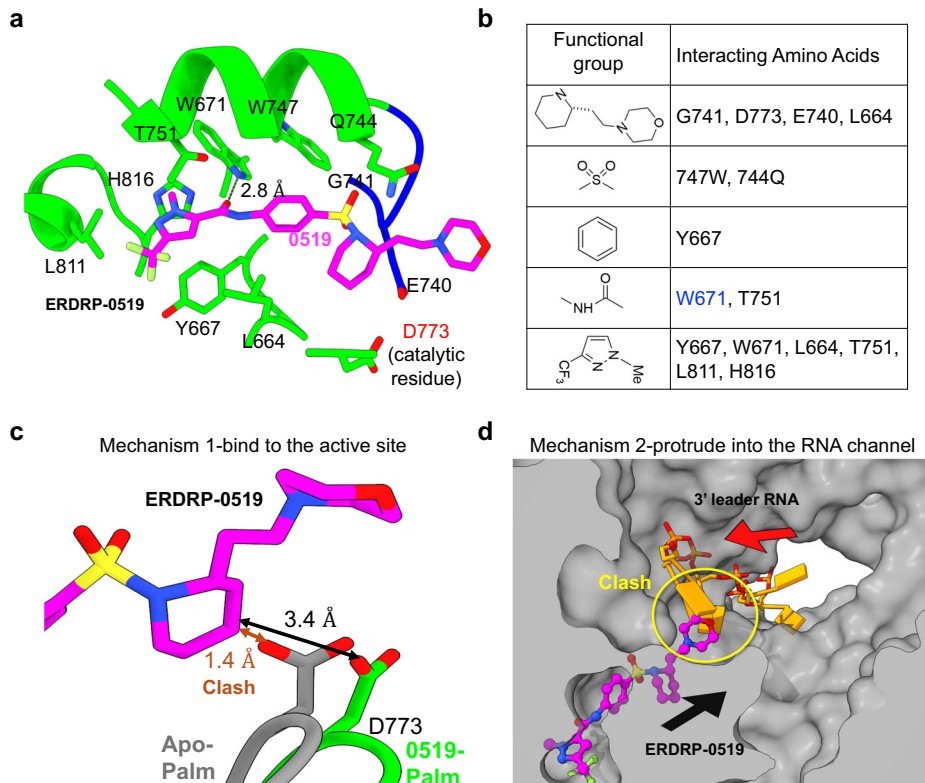

**Fig. 2 | Structural mechanisms of ERDRP-0519 inhibition. a** Interactions between the palm subdomain and ERDRP-0519, with the hydrogen bond to W671 shown as a dashed line. **b** List of residues interacting with ERDRP-0519's functional groups. **c** Superimposition of MeV L$_{full}$-P-C-ERDRP-0519 and apo-MeV L$_{full}$-P-C (PDB: 9DUT), showing D773 displaced outward, forming a hydrophobic interaction with the

compound. **d** Superimposition of MeV L$_{full}$-P-C-ERDRP-0519 and NiV L-P-RNA (PDB: 9GJU), with 3' leader RNA shown as yellow sticks and the clash between ERDRP-0519 and RNA marked. The clip surface of MeV L protein (gray) displays the ERDRP-0519 binding pocket and the potential 3' leader RNA channel.

comparison with the NiV L-P-RNA complex[17] shows steric clashes, suggesting the morpholine ring obstructs RNA binding (Fig. 2d). Thus, ERDRP-0519 likely impairs RNA synthesis through direct engagement of the catalytic center and by inducing allosteric effects.

## Conserved binding and antiviral efficacy of ERDRP-0519

Interestingly, ERDRP-0519 demonstrates broad-spectrum activity against various MeV strains[6,11]. Mapping its contact residues onto a sequence alignment of MeV isolates (Supplementary Fig. 3) reveals high conservation, explaining its consistent antiviral efficacy. These residues are also conserved in Canine distemper virus (CDV), a related Morbillivirus, supporting ERDRP-0519's effectiveness against both MeV and CDV[11]. Extending its antiviral activity to other *Paramyxoviridae* or even *Mononegavirales*, will require further chemical optimization. Sequence comparisons (Fig. 3a) highlight key differences in binding site residues across viral genera, indicating that structural modifications will be necessary for broader activity.

In addition, our structural analysis showed that the linker region of ERDRP-0519 is critical for binding. A strong hydrogen bond (2.8 Å) forms between the compound's carboxyl group in the linker and residue W671 of the L protein (Fig. 2a). To test the importance of W671, we first assessed ERDRP-0519's effect on wild-type MeV polymerase using a cell-based minigenome assay, observing dose-dependent inhibition (Fig. 3b). We then created a W671F mutant polymerase (Supplementary Table 2), disrupting the hydrogen bond since phenylalanine cannot form this interaction. Additional nearby mutations altering hydrophobicity or causing steric hindrance were introduced to evaluate their impact on binding. As shown in Fig. 3c, ERDRP-0519 failed to inhibit the W671F mutant, confirming W671's essential role in binding and antiviral activity. ERDRP-0519 also does not bind to RSV L

protein[13], which has phenylalanine at position 671 (F671), supporting the importance of tryptophan at this site. These findings suggest modifying ERDRP-0519's linker region could expand its antiviral spectrum, especially against viruses lacking W671. Additionally, mutations in the binding pocket impaired polymerase activity, likely due to disruptions near the GDN catalytic site.

Overall, our structural study provides in-depth insights into the binding site and inhibitory mechanism of ERDRP-0519. High-resolution structures of the MeV polymerase complexed with ERDRP-0519 reveal a unique binding pocket within the catalytic GDN motif of the RdRp palm subdomain, distinct from previously predicted sites and known RdRp inhibitors. We demonstrate how key interactions, including a hydrogen bond with W671, underpin compound efficacy and align with resistance mutations observed in functional assays. Our findings also support ERDRP-0519's pan-Morbillivirus activity and offer a framework for future drug modifications to extend its antiviral spectrum beyond Morbillivirus. This study lays a solid foundation for designing next-generation polymerase inhibitors and advancing ERDRP-0519 as a promising therapeutic for measles and related viral diseases.

While our manuscript was under review, a similar study reporting the MeV polymerase structure bound to ERDRP-0519 was published[18]. They also determine the NiV polymerase-ERDRP-0519 complex, highlighting the compound's broader antiviral potential across *Mononegavirales*.

## Methods
### Protein expression and purification

The non-codon-optimized L gene (GenBank: AB052820.1), with an N-terminal twin Strep-tag, and the P gene (GenBank: AB046113.1), with a C-terminal 6×His tag, from the Measles virus strain Edmonston, were

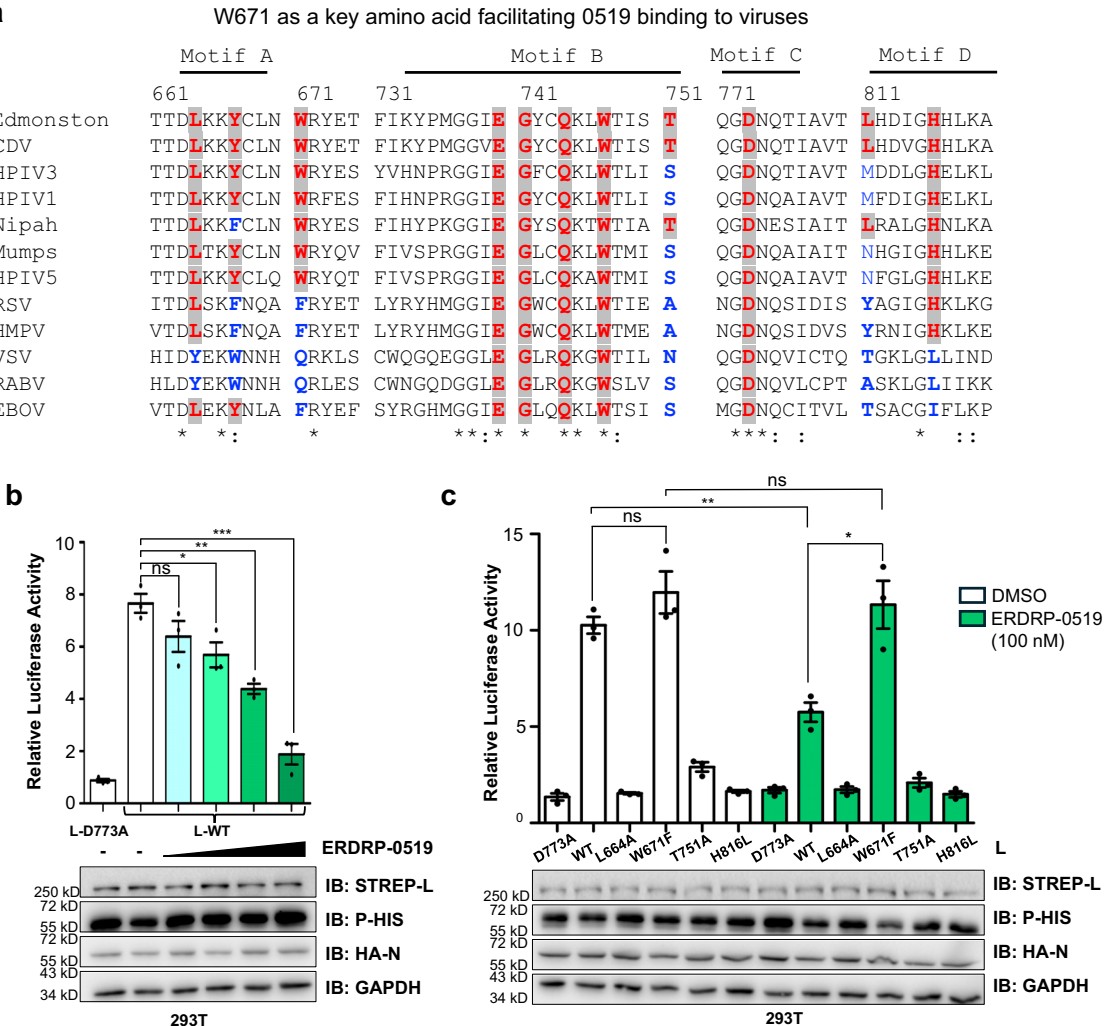

**Fig. 3 | Conservation and functional validation of ERDRP-0519 binding. a** RdRp sequence alignment across *Mononegavirales*. MeV contact residues in red; conserved and non-conserved sites marked red and blue. Sequence sources: MeV (GenBank: K01711.1); CDV (KJ147057.1); HPIV3 (ARA15380.1); HPIV1 (ARB07783.1); NiV (AAY43917.1); MuV (AWI67642.1); HPIV5 (YP_138518.1); RSV (YP_009518860.1); HMPV (Q6WB93.1); VSV (Q98776.1); RABV (AAT48626.1); EBOV (NP_066251.1). **b** Dose-dependent inhibition of MeV polymerase by ERDRP-0519 (0, 1, 10, 100, 1000 nM) using the MeV NanoLuc minigenome assay. **c** Effect of ERDRP-0519 (100 nM) on MeV L mutants assessed by the MeV NanoLuc minigenome system. D773A serves as a negative control. Bars (**b**, **c**) show mean ± SD. Black circles indicate individual values ($n$ = 3). Significance by two-tailed unpaired t-test (*$P ≤ 0.05$, **$P ≤ 0.01$; ***$P ≤ 0.001$; ns, not significant). Source data are provided as a Source Data file.

subcloned into the pFastBac Dual vector and used for the Bac-to-Bac baculovirus expression system, as previously described[14,19]. Briefly, the amplified baculovirus expressing the complexes was used to infect Trichoplusia ni (Tni) cells at a multiplicity of infection (MOI) of 1 at 27 °C for 72 hours. Cells were harvested and resuspended in lysis buffer (25 mM HEPES, pH 7.8, 500 mM NaCl, 5% glycerol, 4 mM $MgCl_2$, 1 mM tris(2-carboxyethyl)phosphine (TCEP), 0.01% Tween 20, 10 mM imidazole), supplemented with cOmplete™ EDTA-free protease inhibitor cocktail (Roche). Following sonication and centrifugation, the lysate was loaded onto a HisTrap column pre-equilibrated with buffer A (25 mM HEPES, pH 7.8, 500 mM NaCl, 5% glycerol, 1 mM TCEP, 4 mM $MgCl_2$). The bound protein was eluted with buffer B (buffer A supplemented with 500 mM imidazole). Eluted fractions were pooled and incubated with Strep-Tactin XT resin (IBA, 2-5030-010) for 30 minutes, followed by four washes with wash buffer (25 mM HEPES, pH 7.8, 500 mM NaCl, 5% glycerol, 1 mM TCEP, 4 mM $MgCl_2$). The protein was then eluted with elution buffer (wash buffer supplemented with 50 mM biotin). Further purification was carried out by size-exclusion chromatography (Superose 6 Increase 10/300 GL, GE Healthcare) using a buffer containing 25 mM HEPES, pH 7.8, 500 mM NaCl, 5%

glycerol, 1 mM TCEP, and 4 mM $MgCl_2$. Fractions near the peak maximum were analyzed by SDS−PAGE, and those containing the MeV polymerase complex were pooled and concentrated to 1 mg/mL. The final sample was flash-frozen and stored at −80 °C.

**Cryo-EM sample preparation and data acquisition**
The purified MeV RNA polymerase complex was diluted 1:1 with the buffer (50 mM HEPES pH 7.8, 50 mM NaCl, 1 mM TCEP) containing 50 μM ERDRP-0519 (AOBIOUS INC, Cat No. AOB2063) and incubated on ice for one hour. 4 μl mixture was applied to freshly glow-discharged Quantifoil R1.2/1.3 300-mesh copper grids (EM Sciences) and blotted for 4 seconds at 4 °C under 100% chamber humidity and plunge-frozen in liquid ethane using a Vitrobot Mark IV (FEI). Images were collected using the K3 Summit detector (Gatan) in super-resolution mode with binning 2 and CDS, along with a Gatan Bio-Continuum GIF energy filter (slit width of 20 eV) at the Hormel Institute, University of Minnesota[20]. The data collection was performed using the EPU software (Thermo Fisher Scientific) with a pixel size of 0.664 Å (nominal magnification of 130,000×) and a nominal defocus value between −1.0 to −2.0 μm. Each image consists of 40 dose-framed

fractions and was recorded with a total dose of 50 e$^-$/Å$^2$. Cryo-EM data collection statistics are summarized in Supplementary Table 1.

## Image processing

Cryo-EM data were processed using cryoSPARC v4.5.1[21], and the detailed procedure is outlined in Supplementary Fig. 1. Briefly, a total of 8,046 dose-fractionated movies were subjected to Patch motion correction using MotionCor2[22] and Patch CTF estimation using CTFFIND-4.1.13[23], with data downsampled by 3/4 (0.885333 Å/pixel after downsampling). Images with the defocus values outside of −0.5 to −3.2 μm or CTF fit resolutions worse than 7 Å were excluded from further steps. A total of 4,333,399 particles were then picked using both the Blob picker and then Template picker in cryoSPARC v4.5.1 and subjected to the Remove Duplicate Particles Tool. Junk particles were removed through one round of 2D classification. A total of 1,325,963 particles from the good 2D classes were used for Ab-initio reconstruction of four maps. The heterogeneous refinement produced 524,675 and 358,252 particles from the good classes, which were then subjected to further non-uniform and CTF refinements to generate final maps of 2.48 Å and 2.73 Å resolution, respectively. Map resolution was determined by gold-standard Fourier shell correlation (FSC) at 0.143 between the two half-maps. Local resolution variation was estimated from the two half-maps in cryoSPARC v4.5.1.

## Model building and refinement

The MeV polymerase machinery model was docked using our previously determined MeV L-P and L-P-C models (PDB codes: 9DUS and 9DUT), and then manually rebuilt in COOT-0.8.9[24]. The cryo-EM densities were sufficient to define ERDRP-0519. Real-space refinements in Phenix-1.16[25] were performed to obtain the final models. In the real-space refinement, minimization global, local grid search, and adp were performed with the secondary structure, rotamer, and Ramachandran restraints applied throughout the entire refinement. The stereochemistry of all structural models was evaluated using MolProbity[26] in Phenix. Model and map statistics are summarized in Supplementary Table 1. Figures were generated using UCSF Chimera X v1.7[27].

## Minigenome assay

The MeV minigenome encoding NanoLuc was constructed with the NanoLuc gene flanked by a T7 promoter and the 5′- and 3′-terminal untranslated regions of the MeV genome, as described previously[14]. 293T cells were seeded in 24-well plates and transfected with the minigenome system plasmids, including 100 ng pCAGGS-minigenome, 40 ng pCAGGS-T7, 40 ng pCAGGS-N, 40 ng pCAGGS-P, 450 ng pCAGGS-L or L mutants, and 1 ng pGL4.54-luc (firefly luciferase control), using Lipofectamine 3000 (Thermo Fisher Scientific, L3000001). The compound ERDRP-0519 was diluted in DMSO and added to the culture medium 12 hours post-transfection at the indicated concentrations. A consistent concentration of DMSO was maintained across all transfection reactions. L mutations were introduced via site-directed mutagenesis, with primers listed in Supplementary Table 2. At 48 hours post-transfection, cells were lysed with Passive Lysis Buffer (Promega, E1941) for 15 minutes at room temperature. The lysates were centrifuged at 13,000 × g for 10 minutes. For Luciferase assays, 40 μL of lysate supernatant was measured using the Nano-Glo Dual-Luciferase Reporter Assay (Promega, N1610) according to the manufacturer's instructions. Luciferase activity was calculated as the NanoLuc/Firefly luciferase ratio. For Western blot analysis, the lysate supernatant was denatured in 2× sample loading buffer, boiled at 100 °C for 5 minutes, separated by SDS-PAGE, and transferred to polyvinylidene fluoride (PVDF) membranes (Millipore, IPVH00010). Membranes were blocked with 5% skim milk in Tris-buffered saline containing 0.1% Tween 20 (TBST) for 1 h, followed by incubation with primary antibodies at 4 °C overnight: rabbit anti-HA (1:2000, Cell Signaling Technology, C29F4), mouse anti-Flag (1:2000,

Sigma, F1804), mouse anti-His (1:3000, Invitrogen, His.H8, MA1-21315), and mouse anti-Strep (1:1000, IBA, 2-1507-001). Detection of the primary antibodies was carried out using HRP-conjugated secondary antibodies (Cell Signaling Technology), including goat anti-rabbit (7074S) and rabbit anti-mouse (58802S), each at 1:5000. Blot images were captured using a ChemiDoc MP imager (Bio-Rad).

## Statistical analyses

All data were analyzed and generated using GraphPad Prism 5 software. Statistical analysis was performed using student's t-test for comparisons between two groups and one-way analysis of variance (ANOVA) for comparisons among multiple groups. Graphs display the means of the experiments, with error bars representing one standard deviation. Significance is indicated as follows: ns, not significant; $*P \le 0.05$, $**P \le 0.01$, $***P \le 0.001$.

## Sequence alignment analysis

Sequence alignments were performed using online tools, Clustal Omega, https://www.ebi.ac.uk/Tools/msa/clustalo/ [28] and ESPript 3.0 https://espript.ibcp.fr/ESPript/ESPript/index.php [29].

## Reporting summary

Further information on research design is available in the Nature Portfolio Reporting Summary linked to this article.

# Data availability

The atomic coordinate of MeV L$_{core}$-P-ERDRP-0519 and L$_{full}$-P-C-ERDRP-0519 complexes have been deposited in the PDB with accession numbers 9OCF and 9OCE, respectively. The corresponding cryo-EM density maps have been deposited in the Electron Microscopy Data Bank with accession numbers EMD-70313 and EMD-70312, respectively. The NiV L-P-RNA structure used in this study is available in the PDB under accession code 9GJU. The HCV NS5B-Sofosbuvir structure used in this study is available in the PDB under accession code 4WTG. The FMDV RdRp-Ribavirin structure used in this study is available in the PDB under accession code 2E9R. Source data are provided with this paper.

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

## Acknowledgements

We thank the staff at the cryo-EM facility and instrument core facility in the Hormel Institute, University of Minnesota. This work was supported by the funding granted to B.L. from the Hormel Institute, University of Minnesota.

## Author contributions

B.L. and D.W. conceived the project and designed the experiments. D.W. prepared the proteins and performed functional studies. B.L., D.W., and G.Y. conducted cryo-EM data collection, cryo-EM image processing, atomic model building and refinement. F.B., D.W., and B.L. wrote the manuscript.

## Competing interests

The authors declare no competing interests.
