## [Transparent Peer Review file · Nature Communications]

Structural Basis of Measles Virus Polymerase inhibition by nonnucleoside inhibitor ERDRP-0519

Corresponding Author: Dr Bin Liu

Version 0:

Reviewer comments:

Reviewer #1

(Remarks to the Author)

In the manuscript by Wang and colleagues they describe the binding site of the measles virus L-protein with the inhibitor ERDRP-0519. This inhibitor binds in the RdRp domain at a non canonical location within the RdRp active site, in contrast to in silico predictions. The authors observed the binding of the compound in both the context of the RdRp/PRNTase structure and in the full L-protein with two copies of the C-protein bound, observing no changes in the binding mode of the compound. The map density in describing both complexes is of high quality and supports both the protein and binding mode of the compound. Mutation of W671 shows its importance in compound binding.

The paper is well written, generally clearly presented, and the data support the conclusions. My suggestions are as below and focus on the presentation of the data. Due to the brevity of the paper my review is similarly brief. My review focuses on the data and its interpretation, and does not comment on the suitability of the work for the journal.

Figure 1, a-b. I am not sure that it is important to include the development of the F targeting compound or its potency and perhaps this could be removed.

Figure 2, c. PDB code should be included for the Apo structure.

Figure 2 f,g. The text size is too small.

Extended data figure 1, panel d. Text size should be increased.

Extended data figure 2. PDB codes and model colouring should be included in the figure legend. It should be made clear the compound positions have been modelled from other viral polymerase structures, not MeV or another paramyxovirus L-protein.

Reviewer #2

(Remarks to the Author)

This manuscript focuses on a small molecule inhibitor of the measles virus polymerase, ERDRP-0519. A precursor to this inhibitor was initially identified in a screen by the Plemper group and their collaborators at Emory University (White et al., 2007), who then went on to optimize the inhibitor to yield ERDRP-0519 (Ndungo et al., 2012). The Plemper group performed detailed mechanism of inhibition studies showing that ERDRP-0519 inhibits the measles virus polymerase (Yoon et al., 2009; Cox et al., 2021) and specifically inhibits initiation of RNA synthesis at the promoter (Cox et al., 2021). In their work, Cox et al. performed resistance mapping and photoaffinity labeling to identify the ERDRP-0519 target site and then docking analysis. Based on this analysis, Cox et al., predicted that ERDRP-0519 binds to the PRNTase domain and suggested that it blocked necessary domain movement. The current work by Wang et al. advances understanding of the mechanism by which ERDRP-0519 functions by providing a cryo-EM model of the measles virus polymerase in complex with the ERDRP-0519, showing that it binds to the RdRp domain, close to the RdRp active site. They also perform functional studies using a minigenome to show that mutation of W671, which contacts the inhibitor in their structure, provides resistance.

Although the manuscript does advance the field, it has limitations in terms of novelty and significance and in its presentation

and scholarship. I should add that as I am not a structural biologist, I cannot comment on the validity of the structural analysis. Specific criticisms are described below.

1. In terms of novelty, this manuscript does not present a significant advance in the field given that the structure of the measles virus polymerase has already been published, and the EDRDP-0519 inhibitor has already been described and its mechanism of action characterized. Other papers have been published describing the structures of related virus polymerases in complex with small molecule inhibitors (e.g., PMID 37865687; 37337079) and the quality and quantity of novel data in this manuscript does not match those.

2. The findings presented here appear to be at odds with data published by the Plemper group regarding EDRDP-0519 mechanism of action. Wang et al suggest that EDRDP-0519 functions by inhibiting the catalytic center by interacting with residue D773 in the GDN motif (lines 111-112). However, Cox et al., concluded that EDRDP-0519 functions primarily to inhibit RNA synthesis initiation because they did not observe an effect of the inhibitor on the viral transcription gradient. It is possible that residue D773 is positioned differently when the polymerase is in an elongation conformation and/or that EDRDP-0519 cannot bind an elongating polymerase, explaining why EDRDP-0519 does not affect the viral transcription gradient. The manuscript does not contain an explanation for the different conclusions regarding the mechanism of inhibition. In addition, it should be noted that the structure of the Nipah virus polymerase in complex with RNA template and product (Sala et al., 2025) would be a more appropriate comparison than the Ebola virus polymerase in complex with the promoter RNA.

3. Cox et al. identified 10 different L resistance mutations to EDRDP-0519 in either measles virus and/or closely related canine distemper virus. These are not mapped onto the structure described in this manuscript to show their positioning with respect to the bound inhibitor. In addition, the Cox group performed photoaffinity labeling to identify the site of the inhibitor binding. The locations of the bound peptides identified by Cox have not been mapped onto the structure. This is a missed opportunity to show concordance between the different sets of data.

4. The authors have not provided an explanation for why Cox et al. reached a different conclusion regarding the inhibitor binding site.

Version 1:

Reviewer comments:

Reviewer #1

(Remarks to the Author)

I am happy with these changes.

Reviewer #2

(Remarks to the Author)

This paper is a resubmitted manuscript. The authors have responded to the reviewers' comments and made some changes that improve the manuscript. The main criticism of this reviewer related to the limitations in novelty/ significance, given that ERDRP-0519 had already been described as a measles virus inhibitor. In this regard, the manuscript is unchanged, but in the responses to reviewers' comments, the authors note that a similar work on structure of the measles virus polymerase complex with ERDRP-0519 was published in Cell in July, 2025. It is reassuring that another paper that has a similar finding has been published, lending support to the significance of this manuscript. However, it is important to recognize that the Cell paper goes further than this manuscript does. Not only did the authors of the Cell paper describe a structure of the measles virus polymerase-ERDRP-0519 complex, they also determined that although most non-segmented negative strand RNA virus polymerases lack the ERDRP-0519 pocket, a pocket is present in the Nipah virus polymerase. Further the authors of the Cell paper demonstrated binding of ERDRP-0519 to the Nipah virus polymerase, showed that ERDRP-0519 had higher affinity binding than to the measles virus polymerase, and solved the structure of the Nipah virus polymerase-ERDRP-0519 complex, providing an explanation for the higher affinity binding, and showed that ERDRP-0519 inhibited Nipah virus polymerase activity. Thus, a significant finding in the Cell paper was that the measles virus polymerase - ERDRP-0519 structure was successfully used to identify an inhibitor targeting the Nipah virus polymerase complex.

General response to Editor and reviewers:

Thank you very much for the opportunity to share our work and improve our manuscript. We appreciate the valuable feedback provided by you. We have carefully revised our manuscript in light of the reviewers' comments. Regarding the concerns about the perceived discrepancies between our structural findings and prior functional predictions, we have addressed and clarified these points in our detailed point-by-point response below.

In addition, while our manuscript was under review, a similar work on structures of the measles virus polymerase complex with ERDRP-0519 was published in *Cell* on July 7th, 2025 (Wang, Y. et al. Structures of the measles virus polymerase complex with non-nucleoside inhibitors and mechanism of inhibition. *Cell*, 2025, doi:10.1016/j.cell.2025.06.017, PMID: 40628260), which confirmed and supported the binding pocket of ERDRP-0519 reported in our manuscript. We have added a brief paragraph in the revised manuscript to address this newly published work, which are listed below.

Line 157-159: *'While our manuscript was under review, a similar study reported the MeV polymerase structure bound to ERDRP-0519 was published¹⁸. They also determined the NiV polymerase-ERDRP-0519 complex, highlighting the compound's broader antiviral potential across Mononegavirales.'*

Note: The line numbers are counted based on the highlighted-change version.

Response to Reviewers' Comments:

Reviewer 1:

In the manuscript by Wang and colleagues they describe the binding site of the measles virus L-protein with the inhibitor ERDRP-0519. This inhibitor binds in the RdRp domain at a non canonical location within the RdRp active site, in contrast to in silico predictions. The authors observed the binding of the compound in both the context of the RdRp/PRNTase structure and in the full L-protein with two copies of the C-protein bound, observing no changes in the binding more of the compound. The map density in describing both complexes is of high quality and supports both the protein and binding mode of the compound. Mutation of W671 shows its importance in compound binding.

The paper is well written, generally clearly presented, and the data support the conclusions. My suggestions are as below and focus on the presentation of the data. Due to the brevity of the paper my review is similarly brief. My review focuses on the data and its interpretation, and does not comment on the suitability of the work for the journal.

Response: Thank you for the positive and constructive evaluation of our manuscript. We appreciate your recognition of the high quality of the structural data.

Based on your suggestions for improving data presentation, we have revised the relevant figures and figure legends, and we have split the original Figure 2 into two separate figures (Figure 2 and Figure 3) to enhance clarity and presentation (see below).

Specific comments:

1. Figure 1, a-b. I am not sure that it is important to include the development of the F targeting compound or its potency and perhaps this could be removed.

Response: As suggested, we have removed the related information of the F targeting compound in Figure 1, a-b.

2. Figure 2, c. PDB code should be included for the Apo structure.

Response: PDB code for the Apo structure has been provided in the figure legend.

Line 243: ‘...and apo-MeV Lfull-P-C (PDB: 9DUT),...’

3. Figure 2 f,g. The text size is too small.

Response: Thank you for your valuable suggestion. In order to increase the text size, we have split the original Figure 2 into two separate figures (Figure 2 and Figure 3) to enhance clear presentation.

4. Extended data figure 1, panel d. Text size should be increased.

Response: We have re-organized the Supplementary figure 1d and increased both the figure and text size as suggested.

5. Extended data figure 2. PDB codes and model colouring should be included in the figure legend. It should be made clear the compound positions have been modelled from other viral polymerase structures, not MeV or another paramyxovirus L-protein.

Response: Thank you for your suggestion. We have revised the legend of Extended Data Figure 2 to include the PDB codes and model coloring. Additionally, we have updated the corresponding text with two new reference to clearly state that the compound positions were modeled based on other viral polymerase structures, not MeV or other paramyxovirus L proteins.

Line 423-426: ‘*Supplementary Fig. 2: Structural superimposition of HCV NS5B–Sofosbuvir (PDB: 4WTG) and FMDV RdRp–Ribavirin (PDB: 2E9R) onto MeV L- ERDRP-0519. The color scheme of MeV L RdRp domain is the same as in Fig. 1d. Drug molecules are colored as follows: Sofosbuvir (cyan), Ribavirin (orange), and ERDRP-0519 (purple).*’

Line 88-91: ‘*This binding mode differs from FDA-approved nucleoside RdRp inhibitors, such as Sofosbuvir for hepatitis C virus (HCV)¹⁵ and Ribavirin for foot-and-mouth disease virus (FMDV)¹⁶, which simply occupy the central polymerase cavity (Supplementary Fig. 2).*’

New References:

15 Appleby, T. C. et al. Viral replication. Structural basis for RNA replication by the hepatitis C virus polymerase. *Science* 347, 771-775, doi:10.1126/science.1259210 (2015).

16 Ferrer-Orta, C. et al. Sequential structures provide insights into the fidelity of RNA replication. *Proc Natl Acad Sci U S A* 104, 9463-9468, doi:10.1073/pnas.0700518104 (2007).

Reviewer 2:

This manuscript focuses on a small molecule inhibitor of the measles virus polymerase, ERDRP-0519. A precursor to this inhibitor was initially identified in a screen by the Plemper group and their collaborators at Emory University (White et al., 2007), who then went on to optimize the inhibitor to yield ERDRP-0519 (Ndungo et al., 2012). The Plemper group performed detailed mechanism of inhibition studies showing that ERDRP-0519 inhibits the measles virus polymerase (Yoon et al., 2009; Cox et al., 2021) and specifically inhibits initiation of RNA synthesis at the promoter (Cox et al., 2021). In their work, Cox et al. performed resistance mapping and photoaffinity labeling to identify the ERDRP-0519 target site and then docking analysis. Based on this analysis, Cox et al., predicted that ERDRP-0519 binds to the PRNTase domain and suggested that it blocked necessary domain movement. The current work by Wang et al. advances understanding of the mechanism by which ERDRP-0519 functions by providing a cryo-EM model of the measles virus polymerase in complex with the ERDRP-0519, showing that it binds to the RdRp domain, close to the RdRp active site. They also perform functional studies using a minigenome to show that mutation of W671, which contacts the inhibitor in their structure, provides resistance.

Although the manuscript does advance the field, it has limitations in terms of novelty and significance and in its presentation and scholarship. I should add that as I am not a structural biologist, I cannot comment on the validity of the structural analysis. Specific criticisms are described below.

Response: Thanks for your comments and concerns on our paper. Although ERDRP-0519 was identified over a decade ago, its mechanism of inhibition on the measles virus polymerase has remained unclear. Some groups have attempted to infer the binding site through mutagenesis and other functional studies. As noted: 'Cox et al. performed resistance mapping and photoaffinity labeling to identify the ERDRP-0519 target site and then docking analysis. Based on this analysis, Cox et al., predicted that ERDRP-0519 binds to the PRNTase domain and suggested that it blocked necessary domain movement.' While these earlier studies provided important initial insights, it is crucial to recognize that functional approaches are indirect and can be influenced by multiple confounding factors.

In contrast, structural biology offers direct and unambiguous evidence. Our cryo-EM structure clearly visualizes ERDRP-0519 bound near the RdRp active site, providing a definitive understanding of its mechanism of action. In addition, while our manuscript was under review, a similar work on structures of the measles virus polymerase complex with ERDRP-0519 were published in *Cell* on July 7th, 2025 (Wang, Y. et al. Structures of the measles virus polymerase complex with non-nucleoside inhibitors and mechanism of inhibition. *Cell*, 2025, doi:10.1016/j.cell.2025.06.017, PMID: 40628260), which identified the same binding pocket as reported in our structure and independently supports the binding mode.

Below, we respond in detail to each of the specific concerns raised and have revised the manuscript accordingly.

Specific comments

1. In terms of novelty, this manuscript does not present a significant advance in the field given that the structure of the measles virus polymerase has already been published, and the ERDRP-0519 inhibitor has already been described and its mechanism of action characterized. Other papers have been published describing the structures of related virus polymerases in complex with small molecule inhibitors (e.g., PMID 37865687; 37337079) and the quality and quantity of novel data in this manuscript does not match those.

Response: We respectfully disagree with the statement that *‘the ERDRP-0519 inhibitor has already been described and its mechanism of action characterized.’* Previous studies provided **indirect and predictive insights** into the inhibitor’s mechanism, but no direct structural evidence has been available to confirm these models. Our study is the first to directly visualize ERDRP-0519 bound to the measles virus polymerase using cryo-EM, thus offering conclusive mechanistic insights.

Regarding the comparison to prior structural studies (PMID 37865687; 37337079), those works describe inhibitors JNJ-8003 and MRK-1 bound to the RSV polymerase, which belongs to the *Pneumoviridae* family. These inhibitors bind to a less conserved surface pocket within the PRNTase domain, and their mechanism is likely specific to RSV and human metapneumovirus. In contrast, ERDRP-0519 binds to a different and conserved region within the RdRp core, indicating a broader mechanistic relevance for *Mononegavirales*.

2. The findings presented here appear to be at odds with data published by the Plemper group regarding ERDRP-0519 mechanism of action. Wang et al suggest that ERDRP-0519 functions by inhibiting the catalytic center by interacting with residue D773 in the GDN motif (lines 111-112). However, Cox et al., concluded that ERDRP-0519 functions primarily to inhibit RNA synthesis initiation because they did not observe an effect of the inhibitor on the viral transcription gradient. It is possible that residue D773 is positioned differently when the polymerase is in an elongation conformation and/or that ERDRP-0519 cannot bind an elongating polymerase, explaining why ERDRP-0519 does not affect the viral transcription gradient. The manuscript does not contain an explanation for the different conclusions regarding the mechanism of inhibition. In addition, it should be noted that the structure of the Nipah virus polymerase in complex with RNA template and product (Sala et al., 2025) would be a more appropriate comparison than the Ebola virus polymerase in complex with the promoter RNA.

Response: As the reviewer mentioned, we *‘suggest that ERDRP-0519 functions by inhibiting the catalytic center by interacting with residue D773 in the GDN motif (lines 112-113).’* while *‘Cox et al., concluded that ERDRP-0519 functions primarily to inhibit RNA synthesis initiation because they did not observe an effect of the inhibitor on the viral transcription gradient.’* We don’t view our conclusion as being in conflict with the findings of Cox et al. work since both initiation and elongation rely on the RdRp catalytic center. We appreciate the proposed idea about *‘It is possible that residue D773 is positioned differently when the polymerase is in an elongation conformation and/or that ERDRP-0519 cannot bind an elongating polymerase, explaining why ERDRP-0519 does not affect the viral transcription gradient.’*. We have incorporated this point in the revised manuscript. The revised text is shown below.

Line 114-117: *‘It is noted that ERDRP-0519 does not affect the viral transcription gradient¹³, suggesting that residue D773 may be positioned differently when the polymerase is in an elongation conformation and/or that ERDRP-0519 cannot bind an elongating polymerase.’*

And we appreciate the suggestion that the Nipah virus polymerase-RNA complex (Sala et al., 2025) may offer a more appropriate comparison. We have revised the related text and figure legend, which are shown below.

Line 118: *‘structural comparison with the NiV L-P-RNA complex¹⁷ shows steric clashes,...’*

Line 245-248: *‘Superimposition of MeV L_{full}-P-C-ERDRP-0519 and NiV-L-P-RNA (PDB: 9GJU), with 3’ leader RNA shown as yellow sticks and the clash between ERDRP-0519 and RNA marked. The clip surface of MeV L protein (gray) displays the ERDRP-0519 binding pocket and the potential 3’ leader RNA channel.’*

New References:

17 Sala, F. A., Ditter, K., Dybkov, O., Urlaub, H. & Hillen, H. S. Structural basis of Nipah virus RNA synthesis. *Nat Commun* 16, 2261, doi:10.1038/s41467-025-57219-5 (2025).

3. Cox et al. identified 10 different L resistance mutations to EDRDP-0519 in either measles virus and/or closely related canine distemper virus. These are not mapped onto the structure described in this manuscript to show their positioning with respect to the bound inhibitor. In addition, the Cox group performed photoaffinity labeling to identify the site of the inhibitor binding. The locations of the bound peptides identified by Cox have not been mapped onto the structure. This is a missed opportunity to show concordance between the different sets of data.

Response: Thank you for your comments and concerns. As noted above, functional approaches can point to regions important for drug interaction or polymerase function, they cannot directly resolve the position of a small-molecule ligand at atomic detail, which structural biology can do. So, we focused on listing the residues that directly interact with ERDRP-0519 as revealed in our structure. In addition, we discussed mutations near the conserved GDN motif that were previously identified in functional studies and are consistent with our structural findings in **lines 91–93**: ‘Our data show the binding pocket is framed by escape mutations around the GDN motif, consistent with the resistance observed in the L-T751I mutant, which exhibited a 20-fold increase in EC90 values¹¹.’

4. The authors have not provided an explanation for why Cox et al. reached a different conclusion regarding the inhibitor binding site.

Response: Thank you for your comments and concerns. Regarding the explanation for why Cox et al. reached a different conclusion regarding the inhibitor binding site, we discussed the potential reason in **line 93-96**: ‘Mutations in the PRNTase domain also affect drug binding¹³, likely due to their proximity to the central polymerase cavity. The morpholine ring of ERDRP-0519 extends into this cavity, suggesting resistance may arise from structural changes that indirectly influence binding.’ The mutations identified by functional studies, though not located directly at the binding interface in our structure, may affect the conformation of the central polymerase cavity near the ERDRP-0519 binding pocket and thus contribute to an indirect resistance mechanism.

In summary, we appreciate the comments and concerns, which ultimately highlight the novelty and significance of our work: (1) Our structural determination of ERDRP-0519 bound to the MeV RNA polymerase provides direct, high-resolution visualization of the inhibitor at its binding site, offering definitive identification of the pocket on the RdRp domain (also shown in recent published Cell paper) rather than the previously predicted PRNTase domain based only on indirect functional studies. (2) Our study reveals a unique binding pocket within the conserved RdRp domain, distinct from those observed in RSV inhibitor-bound structures, which may have broader mechanistic relevance across *Mononegavirales*. (3) We further performed targeted mutagenesis based on the observed binding interface, identifying key resistance determinants (such as W671), thereby elucidating a potential drug resistance mechanism with direct functional support.

Response to Reviewers' Comments:

Reviewer 1:

I am happy with these changes.

Response: We appreciate your acknowledgment of our efforts and your valuable contribution to the review process.

Reviewer 2:

This paper is a resubmitted manuscript. The authors have responded to the reviewers' comments and made some changes that improve the manuscript. The main criticism of this reviewer related to the limitations in novelty/ significance, given that ERDRP-0519 had already been described as a measles virus inhibitor. In this regard, the manuscript is unchanged, but in the responses to reviewers' comments, the authors note that a similar work on structure of the measles virus polymerase complex with ERDRP-0519 was published in Cell in July, 2025. It is reassuring that another paper that has a similar finding has been published, lending support to the significance of this manuscript. However, it is important to recognize that the Cell paper goes further than this manuscript does. Not only did the authors of the Cell paper describe a structure of the measles virus polymerase-ERDRP-0519 complex, they also determined that although most non-segmented negative strand RNA virus polymerases lack the ERDRP-0519 pocket, a pocket is present in the Nipah virus polymerase. Further the authors of the Cell paper demonstrated binding of ERDRP-0519 to the Nipah virus polymerase, showed that ERDRP-0519 had higher affinity binding than to the measles virus polymerase, and solved the structure of the Nipah virus polymerase-ERDRP-0519 complex, providing an explanation for the higher affinity binding, and showed that ERDRP-0519 inhibited Nipah virus polymerase activity. Thus, a significant finding in the Cell paper was that the measles virus polymerase - ERDRP-0519 structure was successfully used to identify an inhibitor targeting the Nipah virus polymerase complex.

Response: Thank you for your thoughtful comments and valuable contribution to the review process. We acknowledge the important findings in the recent Cell paper, particularly that the MeV polymerase-ERDRP-0519 structure was successfully used to identify an inhibitor targeting the Nipah virus polymerase complex. However, compared with the published Cell paper, our work provides several key advances: 1) Higher resolution: We determined the MeV polymerase-ERDRP-0519 structure at 2.48 Å, compared with 3.4 Å in the Cell paper. This higher resolution allows for more precise visualization of atomic interactions at the drug-polymerase interface. 2) Resistance mechanisms: By using cell-based assay, we combined structural and functional data to show how specific interactions, such as a hydrogen bond with W671, contribute to compound resistance. 3) Broader structural context: We determined the structure of inhibitor binding not only in MeV L-P complex but also in L-P-C complex, giving a broader picture of how ERDRP-0519 engages the polymerase machinery. We hope that, taken together, our work and the Cell study provide a deeper and more comprehensive understanding of ERDRP-0519's antiviral action and its broader potential against *Mononegavirales*.